# Pharmacodynamics of Five Anthraquinones (Aloe-emodin, Emodin, Rhein, Chysophanol, and Physcion) and Reciprocal Pharmacokinetic Interaction in Rats with Cerebral Ischemia

**DOI:** 10.3390/molecules24101898

**Published:** 2019-05-17

**Authors:** Rong-Rong Li, Xue-Fang Liu, Su-Xiang Feng, Sheng-Nan Shu, Pei-Yang Wang, Na Zhang, Jian-Sheng Li, Ling-Bo Qu

**Affiliations:** 1College of Pharmacy, Henan University of Chinese Medicine, Zhengzhou 450046, China; 13838057402@163.com (R.-R.L.); 15038361700@163.com (S.-N.S.); 18236759561@163.com (P.-Y.W.); zhangnahnzy@163.com (N.Z.); 2Collaborative Innovation Center for Respiratory Disease Diagnosis and Treatment & Chinese Medicine Development of Henan Province, Zhengzhou 450046, China; liuxf0213@163.com; 3Zhengzhou Key Laboratory of Chinese Medicine Quality Control and Evaluation, Zhengzhou 450046, China; 4College of Chemistry and Molecular Engineering, Zhengzhou University, Zhengzhou 450001, China; qulingbo@zzu.edu.cn

**Keywords:** anthraquinones, pharmacodynamics, pharmacodynamics, pharmacokinetics, interaction

## Abstract

(1) Background: Rhubarb anthraquinones—a class of components with neuroprotective function—can be used to alleviate cerebral ischemia reperfusion injury. (2) Methods: The three pharmacodynamic indicators are neurological function score, brain water content, and cerebral infarction area; UPLC-MS/MS was used in pharmacokinetic studies to detect plasma concentrations at different time points, and DAS software was used to calculate pharmacokinetic parameters in a noncompartmental model. (3) Results: The results showed that the pharmacodynamics and pharmacokinetics of one of the five anthraquinone aglycones could be modified by the other four anthraquinones, and the degree of interaction between different anthraquinones was different. The chrysophanol group showed the greatest reduction in pharmacodynamic indicators comparing with other four groups where the rats were administered one of the five anthraquinones, and there was no significant difference between the nimodipine group. While the Aloe-emodin + Physcion group showed the most obvious anti-ischemic effect among the groups where the subjects were administered two of the five anthraquinones simultaneously. Emodin, rhein, chrysophanol, and physcion all increase plasma exposure levels of aloe-emodin, while aloe-emodin lower their plasma exposure levels. (4) Conclusions: This experiment provides a certain preclinical basis for the study of anthraquinone aglycones against cerebral ischemia and a theoretical basis for the study of the mechanism of interaction between anthraquinones.

## 1. Introduction

Cerebral ischemic stroke is a major cause of severe acquired disability worldwide and the morbidity approximately account for 80% of global strokes [1,2]. However, the use of a thrombolysis agent is the only FDA-approved method for the treatment of acute ischemic stroke so far and there is a limited therapeutic window within 3 h, after which there is potential risk of hemorrhagic transformation [3,4,5,6]. Thus, alternative therapies are urgently acquired. It is a feasible direction to treat cerebral ischemia by using the active ingredients in Chinese medicine as a supplement and substitute therapy complementary and alternative treatment.

Modern Chinese medicine clinical research [7] shows that the basic pathogenesis of cerebral ischemia is deficiency, fire, wind, phlegm, qi, and blood, while qi deficiency and blood stasis are the main pathological mechanisms of ischemic stroke. Therefore, the treatment of tonifying qi and activating blood as a basic treatment can often receive good results. More and more studies have shown that cerebral ischemia and reperfusion are complex pathophysiological processes, and are inextricably linked to autophagy, apoptosis, oxidative stress, inflammation, and other reactions [8,9]. It has been preliminarily proven that anthraquinones have related pharmacological activities, such as reducing the activation of the NALP_3_ inflammatory response complex [10,11] and the upstream signaling pathway of nuclear factor kappa B (NF-κB), downregulation of inducible NO synthase [12,13] (i NOS), inhibits the massive production of NO, and reduces nerve damage. In addition, anthraquinones have a special effect on anticoagulant therapy in patients with primary brain tumors or secondary brain metastases [14]. In the current studies, rhubarb anthraquinone total aglycones have neuroprotective functions and can be used for the treatment of cerebral ischemic injury.

Chinese medicine places a high value on drug compatibility [15]. The compatibility of Chinese medicines is not the simple addition of the therapeutic effects of drugs; on the contrary, ingredients promote reciprocal absorption and entrance to the lesion site in the body, prolong residence time, increase blood drug concentration, change pharmacokinetic behavior, and synergistically enhance the drug therapeutic effect. Until now, most of the existing reports [16,17,18,19,20] concentrate on the overall pharmacodynamic study of rhubarb anthraquinone total aglycones, including aloe-emodin, emodin, rhein, chrysophanol, physcion, or the pharmacokinetic study of one of the five anthraquinone aglycones, so the pharmacokinetic interaction among the five anthraquinones should be deeply investigated. After all, these five aglycones, whose chemical structural formulas are shown in Appendix A, have the same parent group and may have the same target, resulting in similar effects. For example, published reports have showed that emodin cannot only significantly reduce the expression of tumor necrosis factor-α (TNF-α) protein and IL-1β protein in cerebral ischemic tissue and reduce the inflammatory response [21], but also enhance the activity of CAT in the brain of mice with cerebral ischemia-reperfusion, promote the decomposition of hydrogen peroxide, and reduce the generation of oxygen free radicals. Rhubarb aglycons have antioxidation, antibacterial, anti-inflammatory, and antiapoptosis effects, simultaneously [22,23,24]. These may cause an upgrade in the efficacy of the drug, and may also result in a weakness in treatment outcome due to antagonism between the drugs. Therefore, unambiguous interpretations and evidence about pharmacodynamic effect and pharmacokinetic interaction between the five anthraquinone aglycones were acquired for more effective compatibility.

Our investigation is the first to explore the systematic pharmacodynamics of the five anthraquinones (aloe-emodin, emodin, rhein, chrysophanol, and physcion) and pharmacokinetic interaction of two of the five anthraquinone aglycones in cerebral ischemia-reperfusion model rat. It was found that the experimental groups all showed anticerebral ischemic injury and the treatment intensity was consistent when different drug combinations were administered. On this basis, the pharmacokinetic study was carried out. We used UPLC-MS/MS to determine the plasma concentration of aloe-emodin, emodin, rhein, chrysophanol, and physcion at different time points, to calculate the pharmacokinetic parameters, and analyze the pharmacokinetic characteristics and pharmacokinetic interaction. Then, combining the pharmacodynamic and pharmacokinetic results, the functional mechanisms the anthraquinone glycosides on each other were speculated.

## 2. Results

### 2.1. Pharmacodynamic Results

The neurological function scores, brain water content, and infarct size of the sham operation group, the model group, the positive group, and the 16 experimental groups are shown in Table 1 and Figure 1. Comparing the model group with the sham operation group, the neurological function score and brain tissue water content were significantly increased and the ischemic brain tissue staining showed a large area of white cerebral infarction. The results indicated that the cerebral ischemia model was established successfully and the pharmacodynamic characteristics could be reflected with the three selected pharmacodynamic indicators. Compared with the model groups, the three pharmacodynamic indicators were significantly decreased in the positive group and the Aloe-emodin, Rhein, Chrysophanol, Aloe-emodin + Rhein, and Aloe-emodin + Physcion groups. Besides, the Aloe-emodin + Physcion group, which showed no significant pharmacodynamic differences with the positive group, possessed the strongest efficacy of the 15 experimental groups.

### 2.2. Pharmacokinetics Interaction Studies

According to the guidelines for the bioanalytical validation method issued by the FDA (Center for Drug Evaluation and Research [25]), we conducted methodology validation of selectivity, linearity, LOQ (limit of quantitation), accuracy, precision, extraction recovery, matrix effect, and stability. The methodological verification results are shown in Appendix A. The chromatographic method has proven to be linear (r > 0.997), precise (relative standard deviation RSD < 8%), and accurate (relative error RE ± 6%) in the concentration range of the experiment. The minimum detection limits for aloe-emodin, emodin, rhein, chrysophanol, and physcion were 0.026, 0.098, 0.098, 0.538, and 0.317 ng, respectively. The experimental conditions we selected can detect the blood concentration of five rhubarb aglycones rapidly, stably, exclusively, and high sensitively.

For each animal group the calculated mean plasma concentrations of the anthraquinone aglycones, plotted as a function of time, are shown in Figure 2. At all sampling times, the plasma level of aloe-emodin was increased by the co-administration of emodin, rhein, chrysophanol, or physcion; while plasma levels of emodin, rhein, and chrysophanol decreased when administered with aloe-emodin, emodin, rhein, chrysophanol, and physcion. Moreover, different pharmacokinetic parameters of the five anthraquinone aglycones were calculated using noncompartmental analysis, as presented in Table 2. Significant differences in AUC_0–t_, AUC_0–∞_, AUMC_0–∞_, and CL/F values were observed when comparing the groups in which subjects were administered one of the anthraquinone aglycones (aloe-emodin, emodin, rhein, chrysophanol, or physcion) with that groups where subjects were administered two of the anthraquinone aglycones.

## 3. Discussion

In this experiment, the matrix effect and extraction recovery rate of five rhubarb aglycones and an internal standard were investigated under the conditions that methanol, acetonitrile, and methanol: acetonitrile (1:1) were used as precipitation solvent, respectively. The results showed that when pure methanol was used as a protein removal agent the impurity interference was minimal and the matrix effect were in the range of 75 to 115%; recovery rates were higher than 90%. Next, we compared four mobile phase systems—methanol–water (A), acetonitrile–water (B), methanol–0.1% formic acid water (C), and methanol-3 mmol/mL ammonium acetate (D). It was found that the organic phase acetonitrile did not possess peak shape following to inaccurate integral value; meanwhile methanol could avoid this phenomenon. Formic acid water can improve the detection of the quasi-ion peak [M + H]^+^ of rhubarb aglycone. However, according to the fact that the five aglycones contain a plurality of phenolic hydroxyl groups [17,26], the five anthraquinones are more likely to exhibit a correspondingly strong signal quasi-ion peak [M − H]^−^ in an alkaline environment. Therefore, methanol—3 mmol/mL ammonium acetate (D) was finally selected as the mobile phase system. Referring to published literature [27], the C_max_, t_1/2_, and AUC_0–t_ in thrombotic focal cerebral ischemia (TFCI)-induced rats were almost twice that in normal rats, and the CL value was significantly lower than in normal rats (*p* < 0.05), showing that the pharmacokinetic characteristics of the five anthraquinones were modified under pathological conditions. Thus, in order to match the treatment background, cerebral ischemia reperfusion model rats were selected as the research objects in this experiment. In order to provide some theoretical basis for the drug administration methods of the five drugs (single or combined), our research studied the pharmacokinetic characteristics and pharmacodynamics of the five anthraquinone glycosides in the cerebral ischemia reperfusion model rats and the pharmacokinetic interactions and pharmacodynamics of the five anthraquinone glycosides in the pairwise administration.

Considering the drug concentration–time curve, aloe-emodin exhibits different degrees of absorption double-peaks. This may be because, when administered orally, aloe-emodin was easily combined with glucose in the form of glycoside [16], and when the concentration of aloe-emodin in the blood reduced, a small amount of glycoside was hydrolyzed, so that aloe-emodin has a smaller peak. After all, aloe-emodin still exists in the form of a metabolite [28]. This phenomenon was consistent with previous studies; aloe-emodin met the two-compartment model [16]. The four aglycones (emodin, rhein, chrysophanol, and physcion) all prolonged aloe-emodin peak times, T_max_; increased peak concentrations, C_max_; reduced elimination rates; and extended t_1/2_. The increasing AUC_0–t_ and AUC_0–∞_ showed increasing aloe-emodin plasma exposure level. Therefore, the absorption was increased, the elimination was slowed, and action time was prolonged. In addition, the decrease in apparent volume of distribution may cause the drug to concentrate on the pathological site, reducing the side effects of aloe-emodin. Comparing the effects of the four anthraquinones on the pharmacokinetics of aloe-emodin, physcion possessed the greatest influence on it, while rhein showed the least effect. Aloe-emodin, rhein, chrysophanol, and physcion all reduced the AUC_0–t_ value of Emodin, and the AUC_0–t_ value decreased to 67.1%, 72.1%, 59.7%, and 35.9%, respectively. However, they have different ways of changing the pharmacokinetics of emodin. Aloe-emodin increases the exposure level of emodin by shortening and increasing the apparent volume of distribution, V, while physcion shortens the t_1/2_ and improves the clearance rate. The significantly shortened T_max_ indicated that the absorption of emodin was faster and the effect was rapider with the co-administered with aloe-emodin, rhein, chrysophanol, or physcion. Aloe-emodin, emodin, chrysophanol, and physcion all significantly reduced the AUC_0–t_, AUC_0–∞_, and C_max_ of Rhein and significantly increased T_max_, V, and CL, which suggested that the drug–time curves become gentle and the absorption is slowed down. This result may create the reductive possibility of drug accumulation in a certain part to cause the side effects. After all, the drug′s therapeutic effect or toxic side effects often depend on whether the drug dose or concentration exceeds a certain value. In general, aloe-emodin can decrease the AUC of the other four anthraquinones and increase the clearance rate and apparent distribution volume, indicating that aloe-emodin has an inhibitory effect on their utilization; meanwhile, emodin, rhein, and chrysophanol increase and reduce the absorption of aloe-emodin. A large number of experiments [29,30,31,32] have shown that both emodin and chrysophanol can inhibit the expression of TNF-α, IL-1β, and vascular adhesion molecule (VCAM-1), enhance the activity of antioxidant enzymes, inhibit the excessive production of NO, scavenge free radicals, and inhibit apoptosis after cerebral ischemic injury. Thus, many identical neuroprotective functions have demonstrated that these two components may have the same targets in the treatment of cerebral ischemia and exhibit competitive inhibition, which may also be the cause of chrysophanol can cause emodin performed reduced absorption, accelerated elimination, and extensive distribution. The effects of emodin, rhein, and chrysophanol on their mutual pharmacokinetic characteristics are similar.

Three pharmacodynamic indicators were considered as a whole. The anticerebral ischemia effect of chrysophanol was the most obvious when subjects were administered one of the five anthraquinones, while the effect of physcion was not obvious. According to the literature [33,34], chrysophanol can attenuate nitrosative/oxidative stress injury and inhibit the endoplasmic reticulum stress response induced by ischemia/reperfusion damage in the brain. In our study, chrysophanol significantly reduced brain water content and cerebral infarct size with no significant difference from positive pharmacodynamics. There was no significant difference in the neurological function scores between the physcion and the model group. However, the pharmacodynamic index, water content, and cerebral infarct size values were higher than other anthraquinone groups. The three pharmacodynamic indicators of aloe-emodin + rhein and aloe-emodin + physcion were significantly different from the model group and were statistically significant. Comprehensive evaluation of the values of the three indicators, the neurological function score, brain water content and cerebral infarct size values in the aloe-emodin + physcion group were smaller than other compatibility groups, indicating that the group showed the best effect among the compatibility groups. The three indicators of the aloe-emodin + physcion group were lower than those in the aloe-emodin group and physcion group, but, comparing with the model group, the deviation was not equal to the sum of reduction values in the corresponding groups in which the subjects were administered only one of the two anthraquinones. Aloe-emodin and physcion increased the reciprocal anti-ischemic effect, and their interaction was not a simple additive effect but a synergistic effect. Comparing the group with emodin and the compatibility groups with the combination of emodin and the other four anthraquinones, physcion reduced the two pharmacodynamic indicators values of cerebral water content and cerebral infarction area significantly. Therefore, when emodin was combined with other rhubarb aglycones to treat cerebral ischemic injury, physcion may be the best choose. The three indexes in the chrysophanol group were lower than the chrysophanol compatibility groups, and the cerebral infarction area was significantly different from the chrysophanol compatibility groups, indicating that the anticerebral ischemia effect of chrysophanol was better than the chrysophanol compatibility group.

Combined with pharmacodynamic results and pharmacokinetic results, it was found that the pharmacodynamic intensity of the drugs often related to the plasma concentration of the drug in plasma. In the pharmacokinetic study: the AUC_0–t_ of chrysophanol was significantly decreased under the influence of the other four anthraquinone aglycones. Aloe-emodin, emodin, rhein, and physcion reduced the AUC_0–t_ of chrysophanol by 15.5%, 48.4%, 4.1%, and 33.3%, respectively, and had similar trends with brain water content in pharmacodynamic studies. Brain water content: Emodin + Chrysophanol > Physcion + Chrysophanol > Aloe-emodin + Chrysophanol > Rhein + Chrysophanol > Chrysophanol.

## 4. Materials and Methods

### 4.1. Material

#### 4.1.1. Chemicals and Materials

The reference standards of aloe-emodin, emodin, rhein, chrysophanol, physcion, and 1,8-dihydroxyanthraquinone (internal standard (IS)) were purchased from the National Institutes for Food and Drug Control (Beijing, China). The chemical purity of these reference substances was higher than 98% as determined by HPLC analysis. The raw materials had a purity of ≥ 90%; aloe-emodin, emodin, rhein, chrysophanol, and physcion were obtained from Xi′an Xin Rui Bio Technology Co., Ltd. 2,3,5-triphenyltetrazolium chloride (TTC) was obtained from Beijing Solarbio Science & Technology Co. (Beijing, China), Ltd. (Lot. No. 20160310), and chloral hydrate was provided by the Tianjin Guangfu Fine Chemical Research Institute (Tianjin, China). Nimodipine was purchased from Shandong Xinhua Pharmaceutical Co., Ltd. (Shangdong, China), penicillin sodium for injection was purchased from North China Pharmaceutical Co., Ltd. (Hebei, China), and sodium carboxymethyl cellulose (CMC-Na) was purchased from Tianjin Hengxing Chemical Reagent Manufacturing Co., Ltd. (Tianjin, China). Methanol, provided by Tedia Company (Fairfield, OH, USA), and ammonium acetate, purchased from Fisher Company (Pittsburgh, PA, USA), were of mass spectrum grade. The ultrapure water was produced by the Millipore Milli-Q system (Bedford, MA, USA). When aloe-emodin, emodin, rhein, chrysophanol, and physcion were used, the corresponding concentration suspension was prepared with 0.5% sodium carboxymethyl cellulose.

#### 4.1.2. Animals and Cerebral Ischemia Reperfusion Model in Rats

Male Sprague-Dawley (SD) Rats weighing 260–300g were purchased from the Hebei Province Experimental Animal Center (Hebei, China, SCXK2012-1-003). The study protocol is as described by the Experimental Animal Care and Ethics Committee of the First Affiliated Hospital, Henan University of Chinese Medicine (Henan, China, SYXK (Yu) 2015-0005). Before operation, all rats were exposed to 12 h light/12 h dark, temperature 22 ± 2 °C, relative humidity 50 ± 2% conditions for 7 days to adapt to the environment. All breeding and research on experimental animals strictly abide by the regulations on the administration of experimental animals in Henan province. In this study, we used a method previously known as middle cerebral artery occlusion [35,36,37,38] (MCAO) to model cerebral ischemia reperfusion in rats.

Briefly, after anesthesia using 10% chloral hydrate, we separated the left common carotid artery (CCA), external carotid artery (ECA) and internal carotid artery (ICA). A six-0 nylon suture monofilament with a rounded tip (Beijing, China, Beijing salon Biotechnology Co., Ltd.) was inserted through a small incision on the left common carotid artery and moved forward into the internal carotid artery until small resistance was felt, which signifies blocking the entrance of middle cerebral artery. Then ligate the proximal end of the internal carotid artery, suture the wound, and place the rats in a rat cage with a clean pad. At this point, the brain deficit model has been established. After 2 h, the filament was slowly withdrawn until there is resistance to allow reperfusion. The sham group exposed only the left side of the vessel without the insertion.

### 4.2. Pharmacodynamic Study of Anticerebral Ischemia-Reperfusion Injury

#### 4.2.1. Grouping and Administration

After 7 days of acclimatization, 180 rats were randomly assigned to one of the following 18 groups (n = 10 each); Sham operation group, MCAO model group, nimodipine positive control group, and 15 experimental groups MCAO + (Aloe-emodin; Emodin; Rhein; Chrysophanol; Physcion; Aloe-emodin + Emodin; Aloe-emodin+ Rhein; Aloe-emodin+ Chrysophanol; Aloe-emodin+ Physcion; Emodin + Rhein; Emodin + Chrysophanol; Emodin + Physcion; Rhein + Chrysophanol; Rhein + Physcion; Chrysophanol + Physcion). Drug concentrations were as follows; Aloe-emodin 13.05 mg/kg, Emodin 34.65 mg/kg, Rhein 17.25 mg/kg, Chrysophanol 39.45 mg/kg, Physcion 26.4 mg/kg, and the positive drug nimodipine 10.8 mg/kg. All of the above drugs were given orally with the corresponding concentration of suspension prepared with 0.5%CMC-Na before use. The dosage was calculated at one percent of the rat′s body weight. The sham operation group and the model group were given 0.5%CMC-Na of the corresponding volume. Rats in the experimental groups and the positive group were intragastrically administered drugs once a day for seven consecutive days. The rats were fasted but water-free at 12 h before the last dose, and then a cerebral apoplexy reperfusion model was established 2 h after the last administration.

#### 4.2.2. Measurement and Evaluation

(1) The neurological function of rats was evaluated by Longa method [39,40,41]. Scoring criteria: 0 points: no symptoms of neurological deficit, normal activity; 1 point: cannot extend the opposite front claw completely; 2 points: turn to hemiplegia side when crawling; 3 points: when walking, the body falls to the side of hemiplegia; 4 points: unable to walk spontaneously, loss of consciousness; 5 points: death. One-to-three points were selected as successful samples of cerebral ischemia. (2) The scope and area of infarction after cerebral ischemia reperfusion were determined by TTC staining, and the infarction area of each group was calculated by Image analysis software. After staining, the nonischemic area was rose-red and the infarction area was white. (3) The dry and wet method was used to the measurement of brain water content on cerebral ischemia side. Cerebral ischemia reperfusion can induce significant increase of brain water content and lead to cerebral edema in rats. Moisture content of brain tissue = (wet brain weight − dry brain weight)/wet brain weight × 100%.

### 4.3. Pharmacokinetic Interaction among Anthraquinone Aglycones in Model Rats

#### 4.3.1. Apparatus and Operation Conditions

For the LC separation, an Ultimate3000 HPLC (Thermo Scientific, San Jose, CA, USA) equipped with a quaternary pump, a cooling autosampler, a column temperature compartment, and UV detector were utilized. An XBridgeTMC18 (2.1 mm × 150 mm, 5 μm) was employed at 35 °C. The mobile phase consisted of methanol (A) and 3 mmol/L ammonium acetate (B), which was delivered at a flow rate of 0.3 mL/min. The applied gradient elution is as follows; 30–60% A from 0 to 2 min, 60–95% A from 2 to 10 min, 95–60% A from 10 to 11 min, 60–30% A from 11 to 12 min, and 30% A from 12 to 13 min. The injection volume was 5 mL.

For MS, a Q-Orbitrap mass spectrometer (Thermo Scientific, San Jose, CA, USA) equipped with a heat electrospray ionization (HESI) operated in the full-scan mode. The sheath gas, auxiliary gas, and sweep gas pressure was 35 bar, 10 bar, and 10 bar, respectively. Positive and negative ions were simultaneous scanned and the spray voltage was set at +3.5 kV or −2.8 kV under positive or negative mode. The capillary temperature and auxiliary gas heater temperature were maintained at 350 °C and 200 °C. The full scan ranges were from 150 to 1500 *m*/*z* with resolution of 70,000. The ions to be measured and labeled for quantitative analysis are as follows; aloe-emodin *m*/*z* 269.0455 [M − H]^−^; emodin *m*/*z* 269.0455 [M − H]^−^; rhein *m*/*z* 283.0248 [M − H]^−^; chrysophanol *m*/*z* 253.03.506 [M − H]^−^; and physcion *m*/*z* 283.0612 [M − H]^−^.

#### 4.3.2. Plasma Sample Preparation

Rat plasma samples were directly subjected to protein precipitation [42,43] with methanol. Plasma samples (100 mL) were spiked with 100 mL of IS (0.916 mg/mL) and 200 mL of methanol, vortexed for 15 min, and centrifuged at 15,000 r/min for 20 min, and then the supernatant was transferred to a clean test tube, evaporated to dryness at 40 °C under a flow of nitrogen gas, the residue was reconstituted in 100 mL of methanol–water (30:70, *v*/*v*) and subsequently, a 5 mL aliquot was injected for HPLC-MS/MS measurement. Because the concentration of rhein is too high, it is not within the range of our standard curve linear range. In the actual operation of measuring the blood concentration of rhein, we diluted the plasma samples after the above treatment with methanol at ratio 1:50 (*v*:*v*).

#### 4.3.3. Standard Solution Preparation and Sample Quality Control

Five reference products (aloe-emodin, emodin, rhein, chrysophanol, and physcion) of rhubarb were accurately weighed, and methanol was used as solvent to prepare stock solutions with appropriate concentration. The five stock solutions were moderately mixed to afford a final mixed standard solution. A series of working solutions of these five analytes were freshly prepared by diluting the mixed standard solution with methanol gradiently. 1,8-Dihydroxyanthraquinone, an IS, was prepared at a concentration of 0.916 µg/mL by methanol. Calibration samples were prepared by spiking 100 µL working solutions of the corresponding concentrations; 100 µL methanol and 100 µL of IS solution to 100 µL of blank rat plasma. Then, calibration samples were pretreated according to the “4.3.2 Plasma sample preparation”. Low-, medium-, and high-quality control (QC) samples were also prepared in the same way; the medium concentrations were of aloe-emodin 93.2 ng/mL; emodin 77.6 ng/mL; rhein 82.8 ng/mL; chrysophanol 85.6 ng/mL; and physcion 86.4 ng/mL, and were used to measure the precision and accuracy of our methods. All stock solutions were refrigerated at 4 °C.

#### 4.3.4. Administration and Sample Collection

After seven days of acclimatization, 150 rats were randomly divided into groups of 10 each. And those subjects were fasted but free to drink water for 12 h before building the cerebral ischemia model. The rats were administered when the cerebral ischemia model established for 12 h. The administered composition and dose of the 15-group pharmacokinetic study were the same as those of the 15 experimental groups in the pharmacodynamic study (4.2.1). Blood was collected from the tail vein before and after administration at 0.083, 0.25, 0.75, 1, 2, 4, 6, 8, 12, 24 h, respectively, and collected in the heparin centrifuge tube. The blood was then centrifuged in a refrigerated centrifuge for 20 min (6000 r/min) to obtain plasma. Plasma was stored in a −80 °C refrigerator for cryopreservation.

### 4.4. Statistical Methods

SPSS19.0 analysis software was adopted for data analysis. The results were expressed in terms of means ± standard deviation and one-way ANOVA was used to analyze the data differences between the groups by some standards; *p* < 0.05 indicates statistical significance, *p* < 0.01 indicates significant difference.

## 5. Conclusions

In each group rats were administered with one of the five anthraquinone aglycones; the anticerebral ischemia effect of chrysophanol was the most obvious. In the compatibility groups, the anticerebral ischemia effect of aloe-emodin + physcion group was the most obvious. After co-administration with Aloe-emodin, the anticerebral ischemia effect of emodin, rhein, and chrysophanol was weakened. Therefore, when treating cerebral ischemic injury, it was not recommended to administer aloe-emodin with emodin, rhein, and chrysophanol. The pharmacokinetic interaction between the five anthraquinones aglycones was intricate. Emodin, rhein, chrysophanol, and physcion increased the absorption and slowed the elimination of aloe-emodin, while aloe-emodin reduced the AUC_0–t_ values of the above four anthraquinones. Emodin, chrysophanol, and rhein have a similar pharmacokinetic effect on each other. It was speculated that the three anthraquinones had the same targets in anticerebral ischemia. In this study, the pharmacodynamics and pharmacokinetics of five rhubarb aglycones were systematically studied, and their pharmacodynamics and pharmacokinetic interaction were explored, which provided a theoretical basis for the study of the mechanism of action.

## Figures and Tables

**Figure 1 molecules-24-01898-f001:**
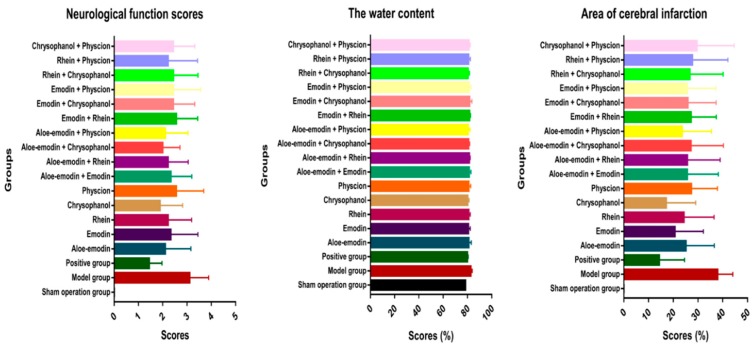
Pharmacodynamic index: the neurological function scores, the water content, and area of cerebral infarction. The results are expressed in terms of means ± standard deviation (n = 10).

**Figure 2 molecules-24-01898-f002:**
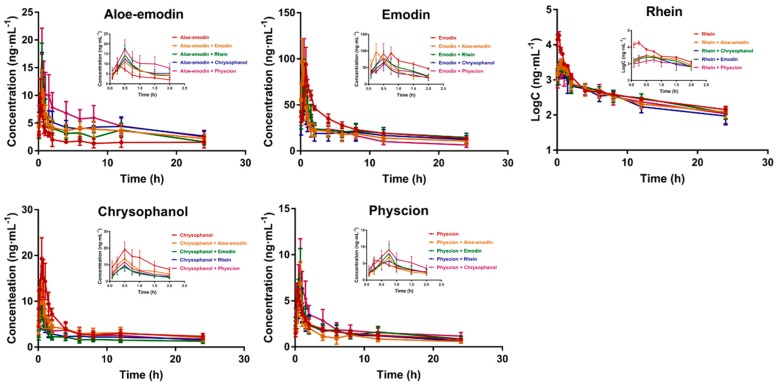
Mean plasma concentration–time curves.

**Table 1 molecules-24-01898-t001:** Pharmacodynamic index: the neurological function scores, the water content, and area of cerebral infarction. The results were expressed as means ± standard deviation. One-way ANOVA and bilateral inspection were used to analyze the data differences between the groups by some standards that ^Δ^, * *p* < 0.05; ^ΔΔ^, ** *p* < 0.01. Δ showed the comparison between the sham operation group, * showed the comparison between the model group.

Group	Neurological Function Score	The Water Content (%)	Area of Cerebral Infarction (%)
Sham operation group	-	78.40 ± 0.28	-
Model group	3.11 ± 0.78 ^ΔΔ^	82.84 ± 1.08 ^ΔΔ^	37.95 ± 6.13 ^ΔΔ^
Positive group	1.44 ± 0.53 **	79.91 ± 0.64 **	14.36 ± 10.20 **
Aloe-emodin	2.11 ± 1.05 *	81.03 ± 2.15 **	25.09 ± 11.47 **
Emodin	2.33 ± 1.12	80.78 ± 1.52 **	20.65 ± 11.51 **
Rhein	2.22 ± 0.97 *	81.08 ± 1.21 **	24.31 ± 12.19 **
Chrysophanol	1.89 ± 0.93 **	80.10 ± 1.21 **	17.16 ± 11.94 **
Physcion	2.56 ± 1.13	81.02 ± 1.64 *	27.32 ± 10.57 *
Aloe-emodin + Emodin	2.33 ± 0.87	81.42 ± 1.45 *	25.62 ± 12.61 *
Aloe-emodin + Rhein	2.22 ± 0.83 *	81.52 ± 0.70 *	25.72 ± 13.21 *
Aloe-emodin + Chrysophanol	2.00 ± 0.71 **	80.91 ± 0.91 **	27.14 ± 13.14
Aloe-emodin + Physcion	2.11 ± 0.93 *	80.52 ± 1.53 **	23.56 ± 11.92 *
Emodin + Rhein	2.56 ± 0.88	81.65 ± 0.91 *	27.16 ± 10.25
Emodin + Chrysophanol	2.44 ± 0.88	81.69 ± 1.98 *	25.91 ± 11.41 *
Emodin + Physcion	2.44 ± 1.13	81.34 ± 1.66 **	25.57 ± 11.63 *
Rhein + Chrysophanol	2.44 ± 1.01	80.38 ± 1.27 **	26.66 ± 13.49 *
Rhein + Physcion	2.22 ± 1.21 *	80.89 ± 1.52 *	27.70 ± 14.38
Chrysophanol + Physcion	2.44 ± 0.88	81.18 ± 1.11 *	29.58 ± 15.04

**Table 2 molecules-24-01898-t002:** Pharmacokinetic parameters of 5 substances in different groups (mean ± SD; n = 10). The pharmacokinetic parameters were calculated according to the blood concentration time of each subject, and then the mean values and SD values of 10 individual pharmacokinetic parameters in each group were calculated. One-way ANOVA and bilateral test were used to compare the pharmacokinetic differences between the compatibility groups and their corresponding groups which the subjects were administered one of the aglycones (* *p* < 0.05; ** *p* < 0.01).

	Group	Parameters								
		t_1/2_ (h)	T_max_ (h)	C_max_ (ng/mL)	AUC_0-t_	AUC_0-∞_	AUMC_0-∞_	MRT_0-∞_ (h)	V/F (L)	Cl/F (L/h)
					(ng·h/mL)	(ng·h/mL)	(ng·h^2^/mL)			
Aloe-emodin	Aloe-emodin	14.98 ± 6.48	0.32 ± 0.12	9.88 ± 2.9	42.77 ± 10.09	63.18 ± 24.20	1524 ± 566.3	21.73 ± 8.35	4287 ± 1948	244.2 ± 118.0
	Aloe-emodin + Emodin	33.26 ± 6.4 **	0.45 ± 0.11	10.85 ± 1.45	88.02 ± 8.32 **	214.4 ± 14.32 **	9303 ± 703.5 **	45.3 ± 6.39 **	2857 ± 171.6 *	59.6 ± 7.76 **
	Aloe-emodin + Rhein	21.51 ± 6.36	0.55 ± 0.11 **	14.1 ± 3.04 *	77.58 ± 7.3 **	126.6 ± 15.53 **	3323 ± 240.7 **	26.61 ± 5.61	3181 ± 271.5	104.0 ± 12.18 **
	Aloe-emodin+ Chrysophanol	28.34 ± 4.72 **	0.45 ± 0.11	12.41 ± 2.53	99.62 ± 11.7 **	210.4 ± 18.37 **	7955 ± 508.8 **	38.15 ± 5.3 **	2488 ± 330.0 **	62.78 ± 5.30 **
	Aloe-emodin + Physcion	15.49 ± 3.64	0.55 ± 0.11 **	17.8 ± 4.18 **	121.5 ± 19.59 **	177.7 ± 16.70 **	3641 ± 261.1 **	21.2 ± 6.57	1556 ± 262.4 **	75.65 ± 15.29 **
Emodin	Emodin	23.13 ± 3.56	0.75 ± 0.00	91.65 ± 16.82	624.7 ± 73.35	1108 ± 191.1	33673 ± 10503	29.90 ± 4.71	1052 ± 119.8	31.99 ± 5.15
	Emodin + Aloe-emodin	26.13 ± 4.44	0.3 ± 0.11 **	95.15 ± 5.47	419.5 ± 21.62 **	798.8 ± 40.85 **	27693 ± 2767	32.5 ± 5.44	1550 ± 113.0 **	44.82 ± 5.12 **
	Emodin + Rhein	28.97 ± 6.17 *	0.45 ± 0.11 **	77.83 ± 6.2 *	511.8 ± 40.91 **	1194 ± 226.7	45531 ± 5933 **	41.11 ± 5.04 **	1282 ± 165.4 **	30.98 ± 4.08
	Emodin + Chrysophanol	26.73 ± 3.18	0.45 ± 0.11 **	65.19 ± 4.96 **	433.4 ± 33.69 **	906.6 ± 29.35 *	33509 ± 3389	38.18 ± 4.87 **	1475 ± 151.6 **	37.92 ± 4.57
	Emodin + Physcion	5.49 ± 0.78 **	0.5 ± 0 **	91.21 ± 8.89	373.3 ± 54.31 **	397.4 ± 82.77 **	3464 ± 655.3 **	8.64 ± 0.83 **	689.7 ± 49.64 **	87.65 ± 7.74 **
Rhein	Rhein	10.20 ± 2.50	0.23 ± 0.06	19290 ± 5420	18866 ± 1739	20893 ± 2262	157175 ± 52352	7.44 ± 1.87	12.08 ± 2.07	0.83 ± 0.09
	Rhein + Aloe-emodin	5.21 ± 0.89 **	0.6 ± 0.29 **	3136 ± 412.6 **	10272 ± 1478 **	10909 ± 1119 **	71400 ± 5075	7.17 ± 0.78	13.51 ± 1.46	1.56 ± 0.53 **
	Rhein + Emodin	9.26 ± 0.87	0.5 ± 0 **	2554 ± 511.0 **	10861 ± 1055 **	11515 ± 1366 **	116864 ± 16106 **	10.76 ± 1.10 **	18.33 ± 1.19 **	1.47 ± 0.55 *
	Rhein + Chrysophanol	8.98 ± 0.74	0.5 ± 0.18 **	3076 ± 299.8 **	8914 ± 535.7 **	10548 ± 1333 **	92427 ± 5960 *	10.07 ± 1.54 **	21.9 ± 2.25 **	1.98 ± 0.28 **
	Rhein + Physcion	10.8 ± 1.12	0.75 ± 0.18 **	1746 ± 102.8 **	8857 ± 845.1 **	10500 ± 1367 **	124300 ± 17430 **	11.42 ± 1.61 **	24.18 ± 2.45 **	1.72 ± 0.54 **
Chrysophanol	Chrysophanol	37.47 ± 14.54	0.54 ± 0.10	19.01 ± 3.93	88.97 ± 10.58	185.2 ± 56.36	8755 ± 3200	42.24 ± 23.55	10372 ± 2880	236.8 ± 97.59
	Chrysophanol + Aloe-emodin	34.61 ± 4.86	0.65 ± 0.11 **	15.22 ± 4.55	75.17 ± 8.9 **	173.6 ± 11.3	7441 ± 368	44.73 ± 5.03	10815 ± 1527	248.6 ± 40.56
	Chrysophanol + Emodin	47.34 ± 11.03	0.62 ± 0.15 **	8.74 ± 2.09 **	45.87 ± 7.42 **	121.8 ± 15.29 **	7125 ± 450	54.53 ± 9.19	20629 ± 2753 **	313.0 ± 42.9 *
	Chrysophanol + Rhein	38.18 ± 5.76	0.72 ± 0.12 **	9.17 ± 1.49 **	52.47 ± 3.07 **	160.2 ± 4.80	8390 ± 418	57.27 ± 6.35	14625 ± 1072 **	241.8 ± 19.13
	Chrysophanol + Physcion	15.75 ± 2.18 **	0.67 ± 0.13 **	11.8 ± 2.36 **	59.38 ± 4.14 **	96.01 ± 11.77 **	2478 ± 345.7 **	19.06 ± 2.38 **	9866 ± 1354	416.4 ± 48.03 **
Physcion	Physcion	30.47 ± 14.4	0.38 ± 0.14	6.24 ± 0.59	36.38 ± 5.58	72.50 ± 41.70	4492 ± 2151	41.50 ± 17.65	13170 ± 6030	433.4 ± 161.6
	Physcion + Aloe-emodin	26.38 ± 4.99	0.75 ± 0.16 **	6.78 ± 0.81	26.71 ± 4.72 **	48.29 ± 6.73 *	1556 ± 220	32.09 ± 5.62	20255 ± 4540 **	545.1 ± 52.62 *
	Physcion + Emodin	16.65 ± 3.17	0.7 ± 0.11 **	8.05 ± 0.56 **	39.51 ± 2.23	60.89 ± 3.57	1359 ± 156.28	22.82 ± 6.99	10336 ± 1131	436.5 ± 38.8
	Physcion + Rhein	12.88 ± 4.28	0.75 ± 0 **	5.82 ± 0.7	34.51 ± 3.92	45.52 ± 4.41 *	759.2 ± 34.69	16.52 ± 2.82	10773 ± 1823	580.0 ± 46.03 **
	Physcion + Chrysophanol	27.49 ± 2.62	0.75±0 **	9.14 ± 0.37 **	48.45 ± 3.47 **	94.39 ± 5.18	3346 ± 260	35.89 ± 4.75	10747 ± 1803	276.7 ± 16.66 **

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
