# Peer review of "Pharmacodynamics of Five Anthraquinones (Aloe-emodin, Emodin, Rhein, Chysophanol, and Physcion) and Reciprocal Pharmacokinetic Interaction in Rats with Cerebral Ischemia"

_molecules, 2019, doi:10.3390/molecules24101898_

Round 1

Reviewer 1 Report

Well done, indeed. My sincere congrats to all authors. Accept after minor revision.

Therapeutic anticoagulation in patients with primary brain tumours or secondary brain metastasis might be also mentioned throughout the text of Your promising manuscript. In such a sense, You may consider citing of the following references:

Nat Prod Res. 2018;32(4):375-384.  

- Curr Top Med Chem. 2013;13(21):2745-2766.

By the way, English itself needs some improvement.

Once again, I am pretty satisfied. Without a doubt, Your valuable manuscript has a real potential to be quite well cited (hetero-citations) in the time to come.

Last but not least, very BEST of (research) luck ahead! 

Author Response

Response to Reviewer Comments  

Point 1: Therapeutic anticoagulation in patients with primary brain tumours or secondary brain metastasis might be also mentioned throughout the text of Your promising manuscript. In such a sense, You may consider citing of the following references:

- Nat Prod Res. 2018;32(4):375-384.  

- Curr Top Med Chem. 2013;13(21):2745-2766.

Response 1:  Thank you for your advice. After careful consideration of your proposal, we cite the relevant literature (- Curr Top Med Chem. 2013;13(21):2745-2766.) in line 57 of the article as our reference number 14. However, it is a pity that the article (-Nat Prod Res. 2018;32(4):375-384.)  was not quoted for low correlation with our research.

Reviewer 2 Report

The authors investigated the drug-drug interactions of 5 substances in a cerebral ischemic rat model in pharmacokinetic and pharmacodynamic points of view. However, the manuscript includes a lot of mistakes, for example in units, therefore, the authors have to revise it from scratch. Although aglycones were used, they mentioned ‘glycosides’. In addition, the authors should interpret and discuss the present results with sound scientific knowledge. This result must not provide the action mechanism of the substances, but just phenotypes.

Previous literatures should be cited and discussed with great care; the authors did not refer to even their work. The followings are some examples.

Feng SX, Li JS, Qu LB, Shi YM, Zhao D. Comparative pharmacokinetics of five rhubarb anthraquinones in normal and thrombotic focal cerebral ischemia-induced rats. Phytother Res. 2013 Oct;27(10):1489-94

Zhao Y, Fang Y, Zhao H, Li J, Duan Y, Shi W, Huang Y, Gao L, Luo Y. Chrysophanol inhibits endoplasmic reticulum stress in cerebral ischemia and reperfusion mice. Eur J Pharmacol. 2018Jan 5;818:1-9

Zhao Y, Huang Y, Fang Y, Zhao H, Shi W, Li J, Duan Y, Sun Y, Gao L, Luo Y. Chrysophanol attenuates nitrosative/oxidative stress injury in a mouse model of focal cerebral ischemia/reperfusion.J Pharmacol Sci. 2018 Sep;138(1):16-22.

The results in Table 1 were duplicated in Figure A, and the meaning of the asterisks has to be shown. There seemed to be no significance compare to model group based on the variations. Statistical methods should be clearly written for significance. What was used for post-hoc analysis?

Pharmacokinetic parameters should be obtained from individuals. However, Table 2 seemed to be listed from the mean plasma concentration-time curves. The units have to be checked as well. The authors have to show how to calculate the parameters exactly.

The resolution of Figure B should be improved.

The substances seemed to be measured by a full scanning with a single mass. However, the author mentioned LC-MS/MS. The authors have to check whether the instrument, the orbitrap mass spectrometry, was reproducible enough to quantify the plasma concentrations. Chromatograms have to be shown.

Units of Supplement Table 1 have to be corrected, and the calibration ranges were improper, especially for rhein. The concentrations for QC samples were also inadequate. Calibration standards should be prepared correctly, e.g., less than 10% of standard solution should be added to drug-free plasma. Moreover, the preparation method for calibration curves should be the same with that for real plasma samples.

In Table S2, accuracy and RSD have to be shown correctly.

The authors have to show how to measure the matrix effect and recovery.

The accuracy of Table S4 has to be checked.

All substances in the middle of the sentences should start with small letters.

Author Response

Response to Reviewer Comments

Point 1: The authors investigated the drug-drug interactions of 5 substances in a cerebral ischemic rat model in pharmacokinetic and pharmacodynamic points of view. However, the manuscript includes a lot of mistakes, for example in units, therefore, the authors have to revise it from scratch. Although aglycones were used, they mentioned ‘glycosides. In addition, the authors should interpret and discuss the present results with sound scientific knowledge. This result must not provide the action mechanism of the substances, but just phenotypes.

Response 1: According to your comments, we have carefully checked and revised the units in the article, and we have also revised some other details. For example, ‘glycosides’ have been changed to “aglycones”.

Point 2: Previous literatures should be cited and discussed with great care; the authors did not refer to even their work. The followings are some examples.

Feng SX, Li JS, Qu LB, Shi YM, Zhao D. Comparative pharmacokinetics of five rhubarb anthraquinones in normal and thrombotic focal cerebral ischemia-induced rats. Phytother Res. 2013 Oct;27(10):1489-94

Zhao Y, Fang Y, Zhao H, Li J, Duan Y, Shi W, Huang Y, Gao L, Luo Y. Chrysophanol inhibits endoplasmic reticulum stress in cerebral ischemia and reperfusion mice. Eur J Pharmacol. 2018Jan 5; 818:1-9

Zhao Y, Huang Y, Fang Y, Zhao H, Shi W, Li J, Duan Y, Sun Y, Gao L, Luo Y. Chrysophanol attenuates nitrosative/oxidative stress injury in a mouse model of focal cerebral ischemia/reperfusion.J Pharmacol Sci. 2018 Sep;138(1):16-22.

Response 2:  Thank you for your effective proposal. Your suggestion added depth to our article. Therefore, we discussed relevant articles in the article and cited several references you recommended in line 151, 199 as reference NO. 27, 34 and 35.

Point 3: The results in Table 1 were duplicated in Figure A, and the meaning of the asterisks has to be shown. There seemed to be no significance compare to model group based on the variations. Statistical methods should be clearly written for significance. What was used for post-hoc analysis?

Response 3: According to your suggestions, we have clearly expressed the statistical method and the significance of the asterisk in the paper in line122-126: “Pharmacodynamic index: the neurological function scores, the water content, and area of cerebral infarction. The results were expressed as means ± standard deviation. And one way ANOVA, bilateral inspection was used to analyze the data differences between the groups by some standards that Δ, *P < 0.05; ΔΔ, **P < 0.01. Δ showed the comparison between the sham operation group, * showed the comparison between the model group.”

Point 4: Pharmacokinetic parameters should be obtained from individuals. However, Table 2 seemed to be listed from the mean plasma concentration-time curves. The units have to be checked as well. The authors have to show how to calculate the parameters exactly.

Response 4: After considering your suggestion, we found that there was a defect in our article. Therefore, related languages in line 131-134 “The pharmacokinetic parameters were calculated according to the blood concentration-time of each subject, and then the mean values and SD values of 10 individual pharmacokinetic parameters in each group were calculated. One-way ANOVA and bilateral test were used to compare the pharmacokinetic differences between the compatibility groups and their corresponding groups which the subjects were administered one of the aglycones (*p < 0.05; **p < 0.01).” for clarifying the calculation process of pharmacokinetic parameters were added. Also, the units in Table 2 were checked as well. For instance, the unit of AUC0-t and AUC 0-∞ have been modified to (ng · h/mL).

Point 5: The resolution of Figure B should be improved.

Response 5: According to your requirement, the modified image has a resolution of 1200.

Point 6: The substances seemed to be measured by a full scanning with a single mass. However, the author mentioned LC-MS/MS. The authors have to check whether the instrument, the orbitrap mass spectrometry, was reproducible enough to quantify the plasma concentrations. Chromatograms have to be shown.

Response 6:  Considering your requirements, we have added relevant chromatogram as a supplement FigreS2 in the paper.

Point 7: Units of Supplement Table 1 have to be corrected, and the calibration ranges were improper, especially for rhein. The concentrations for QC samples were also inadequate. Calibration standards should be prepared correctly, e.g., less than 10% of standard solution should be added to drug-free plasma. Moreover, the preparation method for calibration curves should be the same with that for real plasma samples.

Response 7:  

According to your comment, the units of Supplement Table 1 have been checked.

As for the problem: “the calibration ranges were improper, especially for rhein”, we added our answer in line 321-324 “Because the concentration of rhein is too high, it is not within the range of our standard curve linear range. In the actual operation of measuring the blood concentration of rhein, we diluted the plasma samples after ‘4.3.2 Plasma sample preparation’ with methanol at ratio 1:50 (v: v).”

As for the problem “the preparation method for calibration curves should be the same with that for real plasma samples”, I'm sorry for the misunderstanding caused by our mistake. Nowadays, we have corrected our mistakes in the article. Line 316-321 “Plasma samples (100 mL) were spiked with 100 mL of IS (0.916 mg/mL) and 200 mL of methanol, vortexed for 15 min and centrifuged at 15000 r/min for 20 min and then the supernatant was transferred to a clean test tube, evaporated to dryness at 40 under a flow of nitrogen gas, and then the residue was reconstituted in 100mL of methanol-water (30:70, v/v) and subsequently 5mL aliquot was injected for HPLC-MS/MS measurement.” Line 331-337 “Calibration samples were prepared by spiking 100µL working solutions of corresponding concentrations, 100 µL methanol and 100 µL of IS solution to 100µL of blank rat plasma. Then Calibration samples were then pretreated according to the “4.3.2 Plasma sample preparation”. Low, medium and high-quality control (QC) samples were also prepared in the same way with the medium concentration were of aloe-emodin 93.2ng/mL; emodin 77.6ng/mL; rhein 82.8ng/mL; chrysophanol 85.6ng/mL; physcion 86.4ng/mL to measure the precision and accuracy of our methods.”

Point 8: In Table S2, accuracy and RSD have to be shown correctly.

Response 8:  According to your suggestion, we have carefully examined and modified Table S2. And the revised version was shown in the attached material.

Point 9: The authors have to show how to measure the matrix effect and recovery.

Response 9: According to your comment, we have added the following “According to the preparation method of mixed reference materials, the mixed reference materials with high, medium and low concentrations were prepared respectively. Then the peak areas (A1) of the five aglycones and internal standard were determined respectively according to "4.3.1 Apparatus and operation conditions". Blank plasma samples were spiked with mixed reference at the above high, medium and low concentrations and treated as the “4.3.2 Plasma sample preparation”. Then the peak areas (A2) of the five aglycones and internal standard were determined respectively. The quality control samples with high, medium and low concentrations were directly determined according to the sample treatment method, Then the peak areas (A3) of the five aglycones and internal standard were determined respectively. Each concentration was measured in parallel for 6 times. The matrix effect is the ratio of the peak area (A2) to the peak area (A1) of the corresponding component. The extraction recovery ratio is the ratio of the peak area (A3) to the peak area (A2) of the corresponding component” to supplementary materials.

Point 10: The accuracy of Table S4 has to be checked.

Response 9: According to your comment, we have carefully examined and modified Table S4. And the revised version was shown in the attached material.

Point 11: All substances in the middle of the sentences should start with small letters.

Response 9:  According to your advice, we have started all substances in the middle of the sentences with small letters. However, we did not change the initial letter to lowercase when the substances show the group. For example, “Aloe-emodin + Physcion group” have not been modified.

At the end of the response, thank you for your comments. we have benefited a lot.

Reviewer 3 Report

This manuscript describes about pharmacodynamics and others of 5 anthraquiones.  There may be many interesting results in the manuscript.  However, I think major revision needs before publishing in molecules.  The comments are as follow.

There are many results of anthraquinones as shown in Table 1 and Figure A and B.  However, I think the aim of this study is unclear.  I recommend the authors to write the discussion part more clear.

The structures of 5 anthraquinones should be shown in the manuscript.  They will be useful for understanding the discussion.

Page 9, line 257-; the concentration is different every drugs.  What is the reason?

Author Response

Response to Reviewer Comments

Point 1: There are many results of anthraquinones as shown in Table 1 and Figure A and B.  However, I think the aim of this study is unclear.  I recommend the authors to write the discussion part more clear.

Response 1:  According to your suggestion, we have depicted the aim of this study  more clear in line 156-160 “In order to provide some theoretical basis for the drug administration methods of the five drugs (single or combined), our research studied the pharmacokinetic characteristics and pharmacodynamics of the five anthraquinone glycosides in the cerebral ischemia reperfusion model rats, the pharmacokinetic interactions and pharmacodynamics of the five anthraquinone glycosides in the pairwise administration.”

Point 2: The structures of 5 anthraquinones should be shown in the manuscript.  They will be useful for understanding the discussion.

Response 2: Considering your comment, we added the structures of 5 anthraquinones as Figure S1 in supplementary materials:

Point 3: Page 9, line 257-; the concentration is different every drugs. What is the reason?

Response 3:  The dose of the five anthraquinones was determined by preliminary experiments of pharmacodynamics research, and the concentration ratio of the five components was roughly consistent with the content ratio of the five components in rhubarb.

Round 2

Reviewer 3 Report

I think the revised manuscript is better than the authentic one.